# Expression, activity, and consequences of biochemical inhibition of α- and β-glucosidases in different life stages of *Culex quinquefasciatus*

**Edwin R. Burgess, IV**[1]*, **Neil D. Sanscrainte**[2], **Caitlin E. Taylor**[1], **Lyle J. Buss**[1], **Alden S. Estep**[2]

**1** Entomology and Nematology Department, University of Florida, Gainesville, FL, United States of America,
**2** USDA-ARS Center for Medical, Agricultural, and Veterinary Entomology, Gainesville, FL, United States of America

* edwinburgess@ufl.edu

**Data Availability Statement:** All relevant data are within the manuscript and its Supporting Information files.

## Abstract

Mosquitoes have a wide range of digestive enzymes that enable them to utilize requisite blood and sugar meals for survival and reproduction. Sugar meals, typically derived from plant sources, are critical to maintain energy in both male and female mosquitoes, whereas blood meals are taken only by females to complete oogenesis. Enzymes involved in sugar digestion have been the subject of study for decades but have been limited to a relatively narrow range of mosquito species. The southern house mosquito, *Culex quinquefasciatus*, is of public health importance and seldom considered in these types of studies outside of topics related to *Bacillus sphaericus*, a biocontrol agent that requires interaction with a specific gut-associated α-glucosidase. Here we sought to describe the nature of α-glucosidases and unexplored β-glucosidases that may aid *Cx. quinquefasciatus* larvae in acquiring nutrients from cellulosic sources in their aquatic habitats. Consistent with our hypothesis, we found both α- and β-glucosidase activity in larvae. Interestingly, β-glucosidase activity all but disappeared at the pupal stage and remained low in adults, while α-glucosidase activity remained in the pupal stage and then exceeded larval activity by approximately 1.5-fold. The expression patterns of the putative α- and β-glucosidase genes chosen did not consistently align with observed enzyme activities. When the α-glucosidase inhibitor acarbose was administered to adults, mortality was seen especially in males but also in females after two days of exposure and key energetic storage molecules, glycogen and lipids, were significantly lower than controls. In contrast, administering the β-glucosidase inhibitor conduritol β-epoxide to larvae did not produce mortality even at the highest soluble concentration. Here we provide insights into the importance of α- and β-glucosidases on the survival of *Cx. quinquefasciatus* in their three mobile life stages.

## Introduction

Anautogenous mosquitoes utilize a range of specialized enzymes to process blood meals, including proteases and peptidases [1]. Apyrase (EC 3.6.1.5) is another enzyme important to

**Funding:** The author(s) received no specific funding for this work.

**Competing interests:** The authors have declared that no competing interests exist.

the acquisition of blood meals by mosquitoes. The function of apyrase is to hydrolyze ATP and ADP into AMP and free phosphates, which inhibits platelet aggregation and thus clotting [2]. Blood is only a requirement for egg development in female anautogenous mosquitoes. A female mosquito's remaining nutritional requirements are satisfied by sugar, usually acquired from plant fluids like nectar (reviewed in [3, 4]). Conversely, the male mosquito derives the totality of its nutritional requirements from sugars. The metabolism of these sugar meals requires specialized enzymes to help liberate glucose and even convert non-glucose monosaccharides to glucose from more complex carbohydrates. The glucose made available by these enzymes is utilized for numerous essential physiological processes.

Glycoside hydrolases, including α-glucosidase (EC 3.2.1.20) and β-1,3-glucanase (EC 3.2.1.6) are commonly found in organisms that digest sugars like sucrose and cellulosic tissues, respectively. Previous studies have examined the activities of glycoside hydrolases in different mosquito species and life stages, including salivary α-glucosidases in adult *Ae. aegypti* [5], midgut α-glucosidases in adult *Anopheles aquasalis* [6], and β-1,3-glucanase in the larvae of *Aedes aegypti* [7].

In *Culex quinquefasciatus*, a vector of numerous pathogens of public health importance, α-glucosidase has been a subject of study because of its interaction with *Bacillus sphaericus*, an important larval *Culex* spp. biological control agent. The α-glucosidase '*Cpm1*' is a 60-kDa protein that plays a critical role as the binding site for the binary toxin produced by *Bacillus sphaericus* in larval *Cx. pipiens* [8], with homologs also found in *Cx. quinquefasciatus* and *Anopheles gambiae* ('*Cqm1*' and '*Aqm1*', respectively). This enzyme is expressed in the brush border membranes of the larval midgut. Like so many toxin-based control mechanisms, resistance to *Bacillus sphaericus* has been observed in field strains of *Culex* spp. worldwide [9–11] and is due to different varieties of nucleotide deletions in the α-glucosidase gene [12, 13] or transposons that render the protein incapable of interacting with the binary toxin [14].

Outside of its importance for biological control, there is a paucity of general information on glycoside hydrolase activity in *Cx. quinquefasciatus*, especially information on β-glucosidases, which larvae may use to help liberate beta-linked glucose from cellulosic tissues. Mosquitoes from the *Culex pipiens* complex, including *Cx. quinquefasciatus*, make use of a wide range of aquatic habitats for larval and pupal development, including highly eutrophic areas like storm sewers [15, 16]. We therefore hypothesized that larvae would exhibit both α- and β-glucosidase activity because of the availability of a wide range of nutritional resources, including cellulosic tissues, in their larval habitats. On the other hand, with adults acquiring sugars from primarily plant sources, which contain high concentrations of sucrose and other sugars with alpha-linked glucose molecules, we hypothesized that high activity of α-glucosidases would be seen in adults. The following study describes activity and expression of both α- and β-glucosidases in the three mobile life stages of *Cx. quinquefasciatus*. Biochemical inhibition experiments were conducted to support the involvement of these enzymes in larval and adult survival.

## Methods

### Mosquitoes and chemicals

The USDA-ARS-CMAVE strain of *Culex quinquefasciatus* was used for all experiments. This strain is insecticide susceptible and has been characterized in previous toxicological studies [17–20]. The strain has been reared in mass colony for decades using a standard rearing process. For the experiments described here, egg rafts (12 per pan) provided from the main colony were floated on 3 L of deionized water in a 40 x 60 plastic pan (Del-Tec Products, Greenville, SC) with 0.5 g of dissolved 2:1 alfalfa powder and brewer's yeast as food source. Trays were maintained at 27°C on a 14:10 light dark cycle with provision of an additional 0.5 g of 2:1 food

every other day as a slurry to reduce tray fouling (i.e., adverse microorganism growth). At pupation, pupae were manually picked using a transfer pipette, rinsed with clean tap water, and placed into 30cm x 30cm x 30cm plastic screened cages (MegaScience, Tiawan) for emergence. Collections of larvae, pupae, and adults were made at each life stage. The same rearing and collection procedure was performed at least 3 times using different batches of eggs distinct in time.

The following chemicals were purchased from Fisher Scientific (Waltham, MA, USA): acarbose hydrate (MW = 645.61; >98% purity), conduritol β-epoxide (MW = 162.14; 97% purity), p-nitrophenyl-α-D-glucopyranoside (MW = 301.25; >98% purity), p-nitrophenyl-β-D-glucopyranoside (MW = 301.25; >98% purity).

## Activity of α- and β-glucosidases

Measuring the activity of both α- and β-glucosidases was done similarly to [21]. Separately, one L3 larvae, one-day-old pupae, and one-day-old adult female from the same generation was homogenized in 200 μL citric acid phosphate (CAP) buffer (50 mM citric acid, 50 mM sodium phosphate, 10 mM sodium chloride, pH 6.0) using a bead mill homogenizer. The resulting homogenate was then centrifuged at 10,000 x g and 4°C for 4 minutes, and the supernatant was used as the enzyme source. Solutions of p-nitrophenyl-α-D-glucopyranoside (α-glucosidase) and p-nitrophenyl-β-D-glucopyranoside (β-glucosidase) were made in acetone, diluted to 3.32 mM in CAP buffer, and 180 μL was added to 20 μL of enzyme supernatant (final concentration 3.0 mM, 0.675% acetone) in the wells of a 96-well clear, flat-bottomed plate. The reactions were allowed to incubate at 37°C for 10 min and then stopped by adding 10 μL of 2.5 M NaOH. Measurement of the p-nitrophenoxide chromophore was done at 405 nm in a Bio-Tek Epoch II spectrophotometer (BioTek, Santa Clara, CA, USA) against p-nitrophenol standards (also 0.675% acetone; $R^2$ = 0.975) also stopped with 10 μL of 2.5 M NaOH. Total protein was determined using the Bradford assay, with bovine serum albumin as a standard [22], and then used to normalize the enzymatic activity based on protein concentration (i.e., specific activity). Each biochemical assays was performed on eight individuals from each life stage.

## Expression of α- and β-glucosidases

A portion of the previously mentioned enzyme homogenates was also used for gene expression studies. RNA was purified from 100 μL of the homogenate using a standard silica column spin kit, with a DNAse digestion step to remove residual DNA (ZymoResearch, Irvine, CA), by following the manufacturer instructions for liquid samples. The concentration of RNA was assessed on a NanoDrop8000 (ThermoFisher, Waltham, MA, USA) and samples were stored at -80°C until further use. Reverse transcription using 300 ng of whole purified RNA and oligo dT primers followed manufacturer instructions and reaction times for the SuperScript IV kit (ThermoFisher, Waltham, MA, USA). A no reverse transcriptase reaction was included as a control.

Putative glucosidases were identified by homology search of known glucosidases (EC 3.2.1.20 & EC 3.2.1.21) against the genomes and transcriptomes of *Culex quinquefasciatus* in NCBI and Vectorbase.org. Primers were designed using Primer3 to be specific for transcripts based on data and designed to cover at least one exon-exon junction in every pair to reduce non-specific amplification from contaminating DNA. NCBI lists 4 possible transcripts with β-glucosidase activity (XM_038253477.1-XM_038253480.1) and the primers were designed to target all four using a common region to provide an assessment of global expression level (CPIJ001433_F—CTGCTGTACCCTGTCGAGTC; CPIJ001433_R– TCGTCGCTGAACCACTTCTC). Primers were also designed against a putative *Culex* homolog

of human GH-16, annotated as a b-1,3-glucan binding protein (XM_001845910.2), a glycoside hydrolase known to break down the cell walls of yeasts and fungi, using the recently released Johannesburg assembly in Vectorbase.org (CQUJHB008315_F–AGCATACTTTGGGAGGTG GC; CQUJHB008315_R–CAGCGTTGGACGGATGTAGA). Ribosomal protein L24 was used as the reference gene using previously described primers [23].

Expression studies were conducted on a calibrated QuantStudio6 Flex instrument (ThermoFisher, Waltham, MA, USA) using technical duplicate 10ul reactions in 384-well plates. Each reaction consisted of 5 μL of SYBRSelect (ThermoFisher, Waltham, MA, USA), 0.66 μL of 3 μM F/R primer premix, 3.84 μL of nuclease free water, and 1 μL of cDNA. Samples were amplified using standard "FAST" parameters (95˚C for 1 min, then 40 cycles of 95˚C for 3 sec and 60˚C for 15 sec) and included a final melt curve stage to ensure amplification resulted in a single peak. Amplifications were replicated two or three times. Data were converted to relative expression using the $2^{-\Delta\Delta Ct}$ method [24] against the reference gene and a larval sample as the calibrating sample defined as expression value 1.

### *In vivo* inhibition of α- and β-glucosidases

To measure inhibition of α-glucosidases, mosquitoes were allowed to emerge in the presence of 20% sucrose water for three days (0–3 day old), at which point they were sorted into mixed-sex groups of approximately 27–97 adults under brief $CO_2$ anesthesia and placed in experimental arenas. The arenas were made from 946.4 mL plastic soup containers with lids (Fig 1). Two holes (1.27 cm diameter each) were bored in the arenas, one in the wall approximately 0.95 cm from the bottom, and the other in the lid. A 2 mL microcentrifuge tube, with its cap cut off, was hot glued approximately 3.18 cm from the bottom of the arena to serve as the reservoir for inhibitor plus sucrose treatment or the no inhibitor sucrose control. A piece of blotting paper (0.95 x 3.8 cm) was cut to fit inside the walls of the receptacle, with approximately 1.3 cm extending out of the receptacle to act as a feeding surface for the mosquitoes. The blotting paper was held in place by a small piece of cotton on one side of the receptacle that also acted as a feeding surface. The inhibitor used was acarbose, an inhibitor of α-glucosidases in insects [25, 26]. A 4 mL volume of 10 mM, 1 mM, or 0 mM acarbose in 20% sucrose with blue food coloring (50 mL sucrose solution and 3 drops of blue food coloring; McCormick & Co., Inc. Hunt Valley, MD, USA) was prepared and added to the receptacle until full. Blue food coloring was used to confirm feeding through observation of blue fecal spots deposited in the arenas and blue crops and midguts viewed under a stereoscope. The containers were placed inside an environmental chamber set to 25˚C, 70% RH, and 12:12 light:dark. Dead mosquitoes were collected every 24 h for four days. Mosquitoes that survived the four days were immediately frozen and stored at -80˚C for follow-up measurement of α-glucosidase activity. Each concentration was replicated four times and were temporally blocked by treatment (i.e., 10, 1, and 0 mM treatments were run simultaneously). Follow-up α-glucosidase activity was measured as described on five 4 d old females that had survived to the end of the assays and did not have visibly blue crops or midguts. Prior to glucosidase activity measurement, the right wings were removed from mosquitoes and mounted to glass slides with euparal. Wing length was used as a proxy for body size in subsequent lipid and glycogen determination assays (described below) from these same samples.

Starvation state of the adults used in the α-glucosidase inhibition assay was quantified by measuring lipid and glycogen concentrations using assays adapted from [27–29]. Anthrone reagent was made by adding 75 mL of purified water to a 500 mL Pyrex bottle, quickly adding 190 mL of concentrated sulfuric acid (95–98%), and then immediately adding 375 mg of anthrone (98%). The exothermic reaction of the acid and water combined with careful swirling

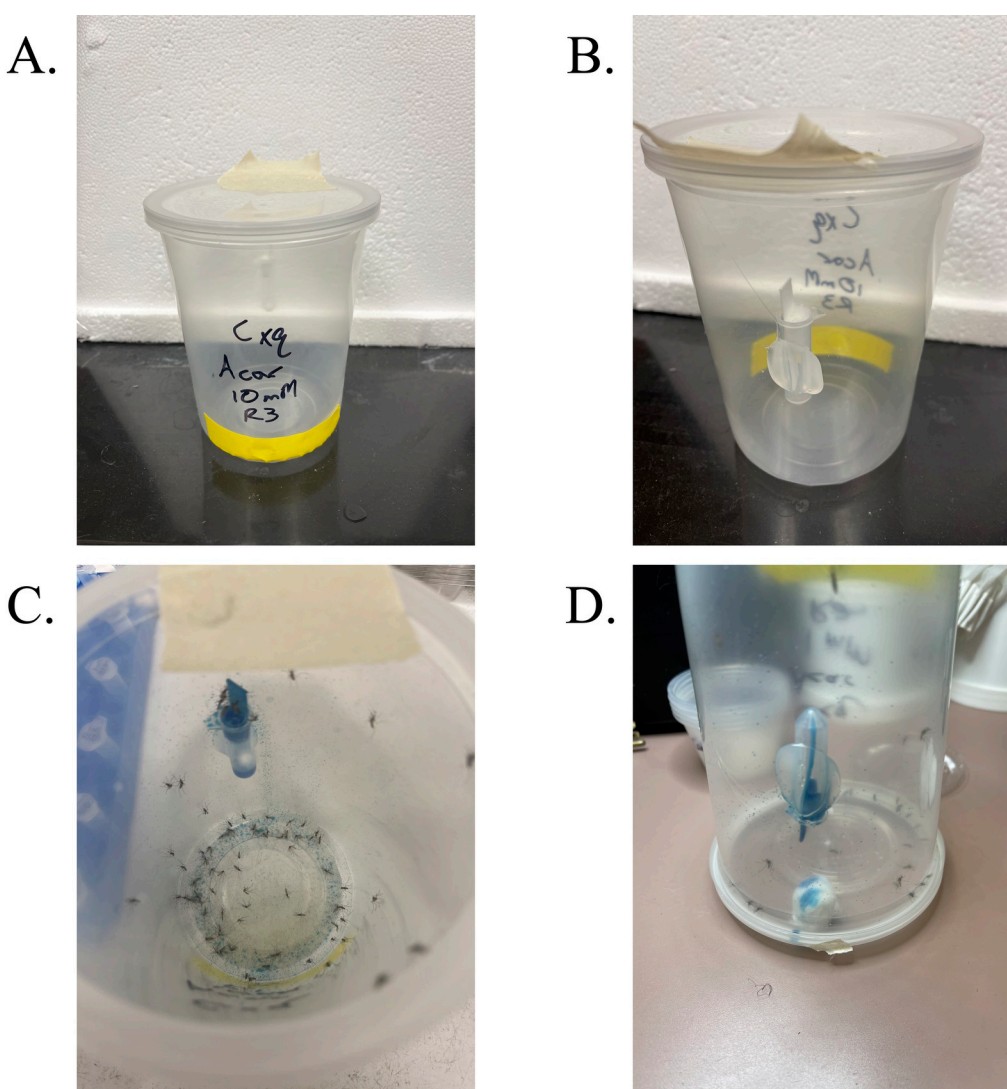

**Fig 1. Diagram of adult test arenas for biochemical inhibition of α-glucosidases in adult *Culex quinquefasciatus*.** A.) the front view of the test arena showing the bottom access hole covered in yellow lab tape, B.) the rear view showing the centrifuge tube feeding reservoir and the top hole covered in masking tape. C.) shows the number of dead mosquitoes on day 4, and D.) shows how the top hole is used to drain the remaining inhibitor in the reservoir before anesthetizing and collecting remaining live mosquitoes for subsequent biochemical tests.

completely dissolves the anthrone. Similarly, vanillin reagent was made by dissolving 350 mg of vanillin (99%) in 50 mL of hot (not boiling) purified water and then mixing it into 200 mL of 85% *O*-phosphoric acid in a 500 mL Pyrex bottle. Both reagents were stored for at least 24 h in light proof containers at 4°C prior to use. After α-glucosidase activity and protein measurement, 140 µL of the supernatant remained. A volume of 60 µL of 6.7% sodium sulfate was added to the supernatant to make a final concentration of 2% sodium sulfate to precipitate glycogen [27]. The supernatant was then mixed with 1000 µL of chloroform:methanol (1:2), vortexed, and centrifuged for 4 min at 10,000 x g and 4°C to pellet the glycogen. The total supernatant was aspirated into 12 x 75 mm borosilicate test tubes and evaporated to complete dryness (~ 1 h) in a Biotage TurboVap LV (Biotage LLC, Uppsala, Sweden) set to 90°C and using building compressed air. Next, 50 µL of concentrated sulfuric acid was added to each

tube, vortexed, and then placed in an aluminum heating block for 10 min at 90˚C. At this point, 1000 μL of anthrone reagent was added to the glycogen tubes, vortexed, and placed in the same aluminum heating block for 8 min. Both the lipid and glycogen tubes were then taken out of the heating block, the anthrone tubes were placed on ice for 10 min to allow for color development, while the lipid tubes had 2000 μL of vanillin added to them, were vortexed, and left to develop at room temp for 30 min. All tubes were vortexed before transferring 100 μL of each tube in duplicate to clear, flat-bottomed, 96-well plates. The lipid samples were read at 525 nm on a spectrophotometer, while glycogen samples were read at 625 nm. Samples were compared to standards of glucose (glycogen) and canola oil (lipids) at 50, 25, 12.5, 6.25, 3.13, 1.56, 0.78, and 0 μg. To account for size differences among the individual mosquitoes, previously removed wings were measured from the apical notch to the axillary margin, excluding the scale fringe on the apical edge of the wing [30, 31] using a Leica M205C stereoscope with attached DMC 5400 digital camera, and Leica Application Suite X measurement software version 3.7.4.23463 (Leica Microsystems, Wetzlar, Germany). Wing length is considered a good proxy for body size when quantifying both glycogen and lipids with the anthrone and vanillin methods, respectively [29].

Because larvae of *Cx. quinquefasciatus* feed on algae, various phytoplankton, and detritus in the water column [32], biochemical inhibition of α- and β-glucosidases was measured. This was done by setting up a series of 70 mL plastic cups with lids, adding 15–25 L3-L4 stage larvae to each cup, along with 0.5 mL of powdered fish food and 0.5 mL of one of four β-glucosidase inhibitor concentrations or a control (total of five cups per replicate). The concentrations tested were 1 mM, 0.1 mM, 0.01 mM, 0.001 mM and 0 mM of conduritol β-epoxide, which is known to inhibit β-glucosidases in insects [33, 34]. Initial mortality experiments were done at these concentrations with high mortality after 24 h due to a buildup of biological film on the surface of the water at all concentrations. Therefore, larvae were collected after a 24 h exposure for quantification of α- and β-glucosidase activity. Formal analysis on larval glucosidase inhibition was not completed as the experiment was terminated after two replicates. Summary statistics are presented in Table 3. A total of five individuals from each inhibitor concentration and replicate (total 10 per inhibitor concentration or control) were measured for both α- and β-glucosidase activity. Activity levels for each concentration were averaged among the five individual larvae tested in each replicate and subtracted from the activity of the 0 mM control to represent the percent remaining activity.

## Statistical analyses

All analyses were conducted with R version 4.2.0 [35]. Activity of α- and β-glucosidases were analyzed by ANOVA of a general linear model, with Box Cox transformation [36] when normality and homogeneity of variance were not met. Post hoc analyses were done pairwise using Tukey's HSD in the 'multcomp' package [37]. The same analysis was used for relative gene expression. Survival data from the adult *in vivo* α-glucosidase inhibition experiments were analyzed with Cox regression using the 'coxme' package [38], including a random intercept assigned to the experimental arenas to account for the lack of independence among individuals measured in each arena (i.e., a "cluster" effect). The resulting measurements of enzyme activity, glycogen, and lipids were analyzed using generalized linear mixed models with the 'glmmTMB' package [39] using a Gaussian family, with activity as the dependent variable and treatment as the independent variable. Replicates were assigned random intercepts to account for the lack of independence of samples within each replicate [40]. Models were initially inspected for statistical assumptions and fit using the 'DHARMa' package [41]. If assumptions or fit were not met, a log transformation improved the models. Global tests of significance

**Table 1. Specific activities of α- and β-glucosidase in L3 stage larvae, 1-day old pupae, and 1-day old adult female *Culex quinquefasciatus*.**

|  | Specific Activity[1] | % of Larval Activity | Ratio of α-to-β-Glucosidase Activity[2] |
|---|---|---|---|
| *α-glucosidase* |  |  |  |
| Larvae | 91.0 ± 11.33 | 100.0 | 0.76 |
| Pupae | 50.6 ± 4.26 | 55.6 | 8.03 |
| Adults | 146.9 ± 18.54 | 161.4 | 19.59 |
| *β-glucosidase* |  |  |  |
| Larvae | 120.2 ± 17.22 | 100.0 | - |
| Pupae | 6.3 ± 0.60 | 5.2 | - |
| Adults | 7.5 ± 0.69 | 6.2 | - |

[1] Mean ± SEM nmol/min/mg protein of p-nitrophenyl-α-D-glucopyranoside (α-glucosidase substrate) or p-nitrophenyl-β-D-glucopyranoside (β-glucosidase substrate) converted to 4-nitrophenoxide.

[2] Ratio represents the specific activity of α-glucosidase divided by the specific activity of β-glucosidase within each life stage. Ratio = 1 means equal activity, > 1 means greater α-glucosidase activity compared to β-glucosidase activity, and < 1 means less α-glucosidase activity.

were done using ANOVA and post-hoc tests were conducted using the 'emmeans' package [42] with Tukey's method for pairwise comparisons. Cutoff for statistical significance for all tests was set to $\alpha = 0.05$.

## Results

### Activity of α- and β-glucosidases

There was a significant difference in α-glucosidase activity among the life stages (Table 1; F = 23.41, df = 2, 21, P < 0.001). Significant pairwise differences arose between larvae and pupae (Table 2; P = 0.002) as well as pupae and adults (P < 0.001), and between larvae and adults (P = 0.023). For β-glucosidase activity, there was a significant difference among the life stages (Table 1; F = 115.66, df = 2, 21, P < 0.001). There were significant pairwise differences between larvae and pupae (P < 0.001), larvae and adults (P < 0.001), but not pupae and adults (P = 0.377).

### Expression of putative α- and β-glucosidases

There was a significant difference among the life stages in the relative expression of *α*-glucosidase (Table 2; F = 6.29, df = 2, 18, P = 0.008). There was a significant pairwise difference between larvae and adults (P = 0.027) and pupae and adults (P = 0.014), but not larvae and pupae (P = 0.907). For the pooled *β*-glucosidases, there was an overall effect on relative expression based on life stage (F = 17.88, df = 2, 20, P < 0.001). There was a significant pairwise difference between larvae and pupae (P = 0.004) and larvae and adults (P < 0.001) but not pupae

**Table 2. Mean and standard error of the mean relative expression[1] of putative α- and β-glucosidases in three life stages[2] of *Culex quinquefasciatus*.**

| Putative target | Larvae | Pupae | Adults |
|---|---|---|---|
| *α-glucosidase* | 1.31 ± 0.492 | 2.05 ± 1.195 | 0.27 ± 0.051 |
| *Pooled β-glucosidases* | 1.29 ± 0.115 | 1.91 ± 0.118 | 2.24 ± 0.118 |
| *β-1,3-glucanase* | 0.85 ± 0.160 | 0.58 ± 0.113 | 0.59 ± 0.051 |

[1] Relative expression was calculated using the $2^{-\Delta\Delta Ct}$ method of Livak and Schmittgen (2001) with ribosomal protein L24 transcript as the reference gene and a larval sample as the calibrator.

[2] Expression was determined from the same samples used to generate Table 1.

**Table 3.** *In vivo* inhibition of α- and β-glucosidases in L4 larval *Culex quinquefasciatus* using no-choice exposure to conduritol β-epoxide.

| Concentration conduritol β-epoxide (mM) | α-glucosidase activity[1] | β-glucosidase activity[1] |
|---|---|---|
| | (% remaining activity[2]) | (% remaining activity[2]) |
| 1.0 | 29.7 ± 4.55 | 41.3 ± 5.24 |
| | (52.0) | (72.1) |
| 0.1 | 37.7 ± 6.35 | 60.0 ± 11.16 |
| | (66.0) | (104.7) |
| 0.01 | 51.3 ± 6.52 | 59.8 ± 6.20 |
| | (89.8) | (104.4) |
| 0.001 | 47.8 ± 3.58 | 49.2 ± 3.59 |
| | (83.7) | (85.9) |
| 0.0 | 57.1 ± 5.01 | 57.3 ± 3.06 |
| | (100) | (100) |

[1] Mean ± SEM nmol/min/mg protein of p-nitrophenyl-α-D-glucopyranoside (α-glucosidase substrate) or p-nitrophenyl-β-D-glucopyranoside (β-glucosidase substrate) converted to 4-nitrophenoxide.
[2] % remaining activity = activity of each conduritol β-epoxide concentration divided by the 0.0 mM control.

and adults (P = 0.137). There was no effect of life stage on the relative expression of β-1,3-glucanase (F = 1.68, df = 2, 21, P = 0.211).

## Biochemical inhibition of α- and β-glucosidases

Adult *Cx. quinquefasciatus* had a 28.1-fold increased mortality rate in the 10 mM acarbose arenas compared to the 0 mM control (Fig 2; HR = 28.1 (14.67–53.94) 95% CI; P < 0.001), and a 15.3-fold increased mortality rate compared to the 1 mM arenas (HR = 15.3 (9.13–25.60) 95% CI; P < 0.001), but no difference was seen between 1 mM and 0 mM acarbose arenas (P = 0.126). Across all treatments, males had a 3.9-fold increased mortality rate compared to females (HR = 3.9 (2.76–5.40) 95% CI; P < 0.001).

There was a significant difference in α-glucosidase activity among the two acarbose treatments and control (Fig 3; $\chi^2$ = 655.6, df = 2, P < 0.001). All pairwise comparisons among the treatments and treatments and control were significantly different (all P < 0.001). Similarly, there was a significant difference in glycogen quantity among the treatments and control (Fig 4; $\chi^2$ = 43.3, df = 2, P < 0.001). The 10 mM acarbose treatment was significantly different from both the 0 mM and 1 mM arenas (both P < 0.001) but there was no difference between 0 mM and 1 mM (P = 0.119). There was also a significant difference in the amount of lipids present among the three treatments (Fig 5; $\chi^2$ = 85.6, df = 2, P < 0.001). Both 10 mM and 1 mM treatments significantly differed from the 0 mM control (both P < 0.001) but were not significantly different compared to each other (P = 0.153). Mean wing length was 3.03 mm ± 0.017.

The *in vivo* inhibition of α- and β-glucosidases with conduritol β-epoxide in larvae never reached the 50% necessary to accurately calculate $IC_{50}$ values (Table 3). Conduritol β-epoxide more strongly inhibited α-glucosidases compared to β-glucosidases (> 20% difference) at 1 mM. This trend was also observed at the other inhibitor concentrations.

## Discussion

The present study suggests that α- and β-glucosidases are critical to the survival of *Cx. quinquefasciatus* and that their roles may be of ontogenetic importance. Clear patterns of activity were observed toward the model substrates p-nitrophenyl-α-D-glucopyranoside (α-

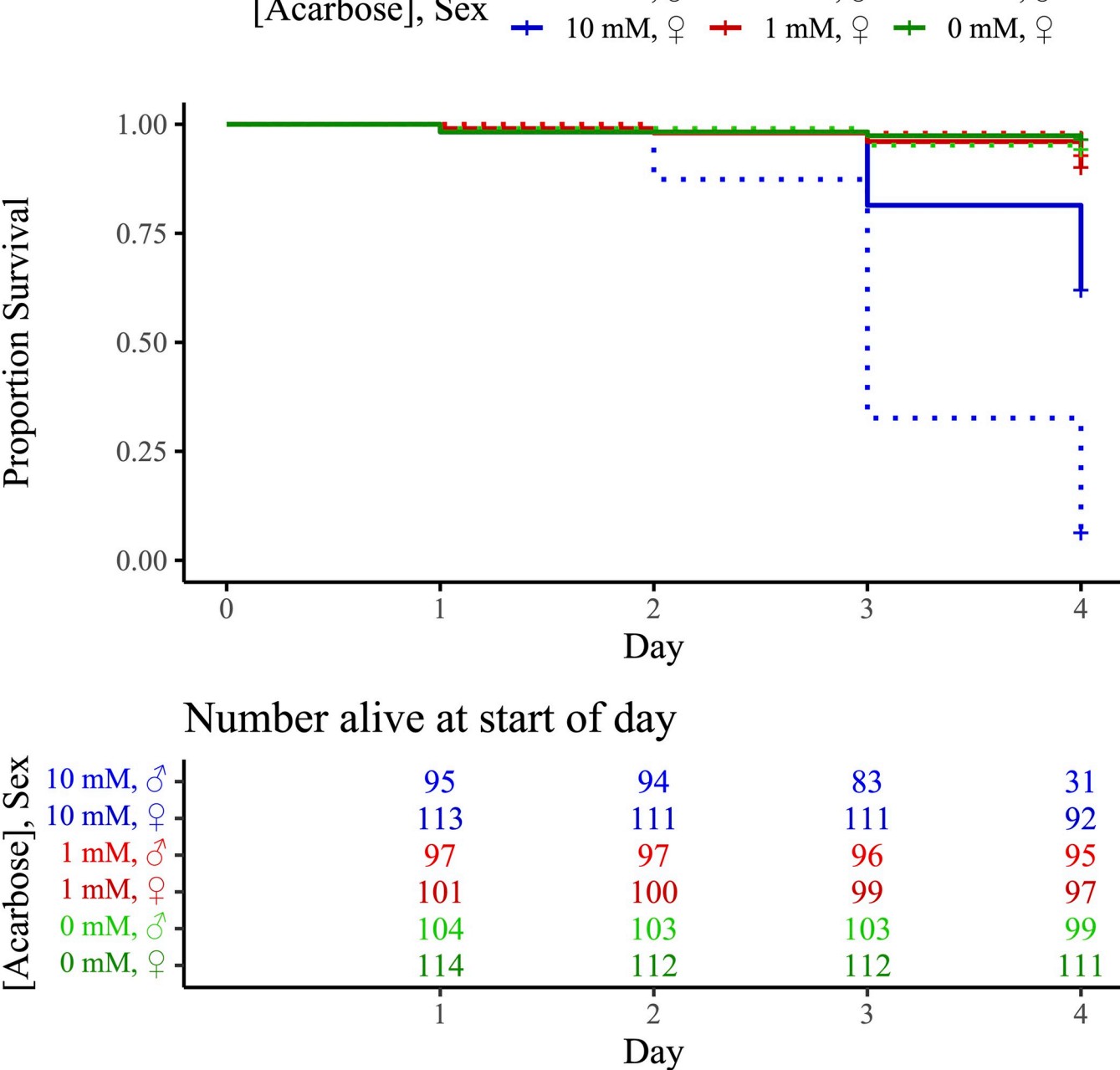

**Fig 2. Survival of adult *Culex quinquefasciatus* in no-choice feeding assays with 10, 1, or mM acarbose (α-glucosidase inhibitor) in 20% sucrose solution.**

glucosidases) and p-nitrophenyl-β-D-glucopyranoside (β-glucosidases) and are likely explained by the carbohydrate resources available at each life stage. While larval α-glucosidase activity was only about 76% that of its β-glucosidase activity, it jumped to over 19-fold of the β-glucosidase activity in adults. This is likely because larval *Cx. quinquefasciatus* are often found in eutrophic aquatic habitats, with abundant cellulosic microorganisms and detritus available, where they employ a collecting-filtering feeding strategy in the water column [43]. These types of carbohydrate sources explain the necessity for β-glucosidases. The α-glucosidase activity we observed in larvae is a little harder to rationalize but may be necessary to

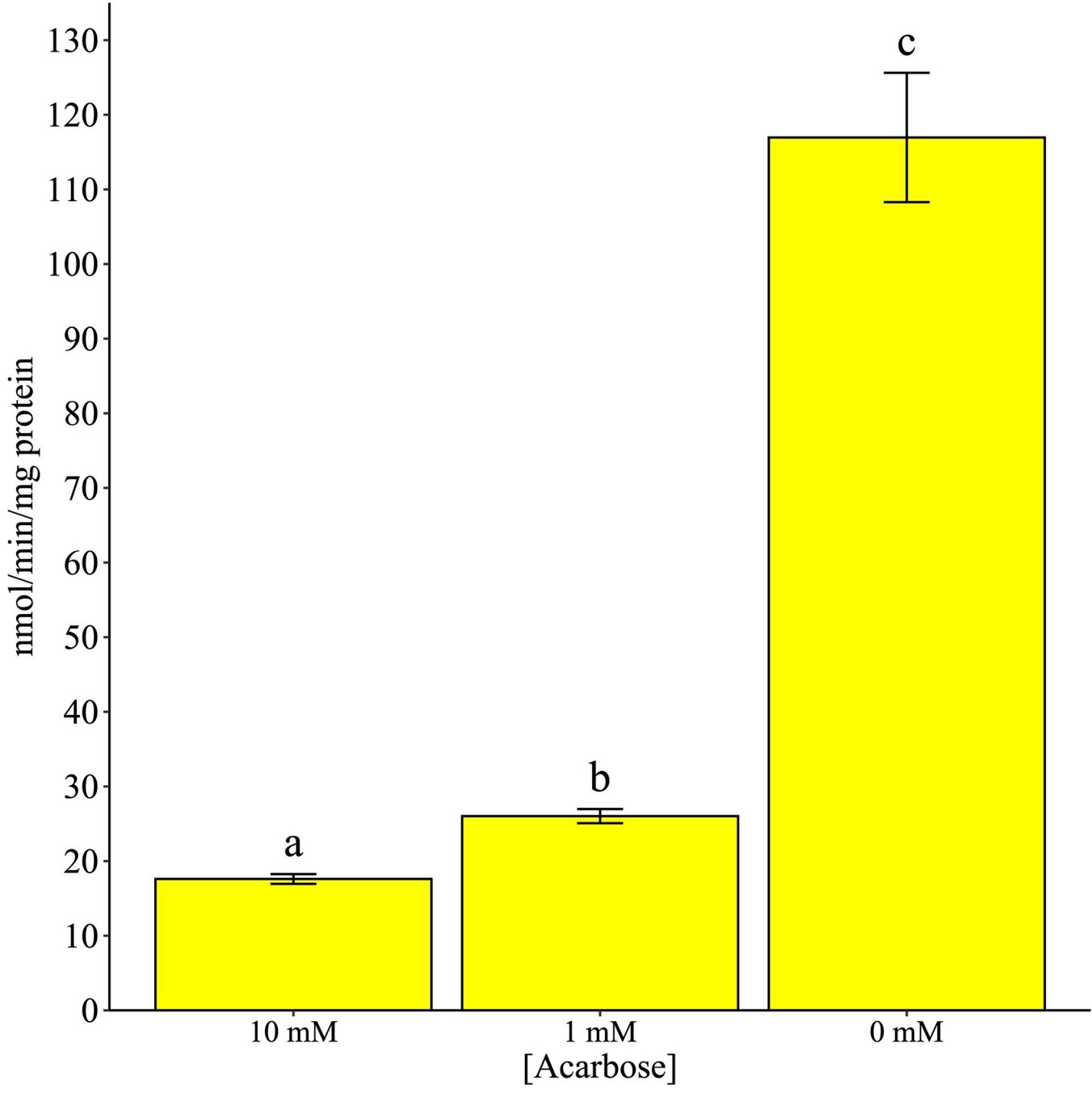

**Fig 3. *In vivo* inhibition of α-glucosidase in adult *Culex quinquefasciatus* using no-choice exposure to acarbose.** Activity was measured by intensity of 4-nitrophenoxide, a yellow chromophore. Different lowercase letters signify statistical significance of pairwise comparisons using Tukey's method at α = 0.05.

process intracellular sugars derived from photosynthetic algaes, which are a major component of most larval mosquito diets (e.g., [44]). Shortly after pupation, a disparity was evident between the two glucosidase activities. While α-glucosidase activity maintained a little over 50% the larval activity, β-glucosidase activity almost completely disappeared. From a developmental standpoint, this latent α-glucosidase activity may be a sort of "priming" scenario, where the pupae are preparing for adult eclosion, or perhaps these enzymes are required for

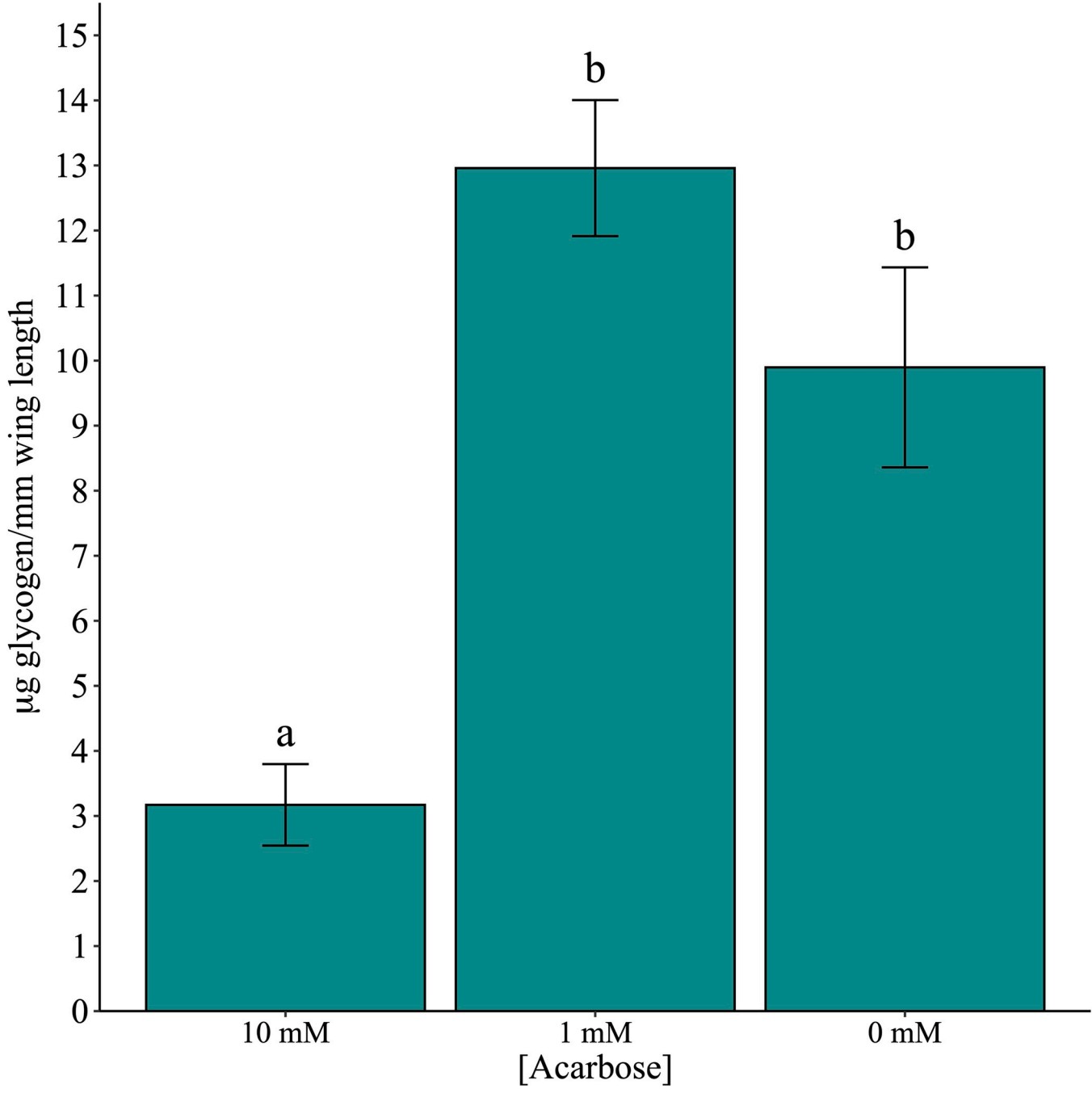

**Fig 4. Glycogen measurements in adult *Culex quinquefasciatus* after no-choice exposure to acarbose using the hot anthrone method.** Glycogen concentration was determined by blue-green color development. Different lowercase letters signify statistical significance of pairwise comparisons using Tukey's method at α = 0.05.

metabolic or catabolic purposes as tissues are reorganized in the pupae. Because adult mosquitoes get their sugar primarily from plant liquids, where sucrose and other glucose-containing saccharides are the main constituents, may explain why adult α-glucosidase activity was over 1.5-fold that of the larvae, while β-glucosidase activity remained at similarly low levels seen in the pupae.

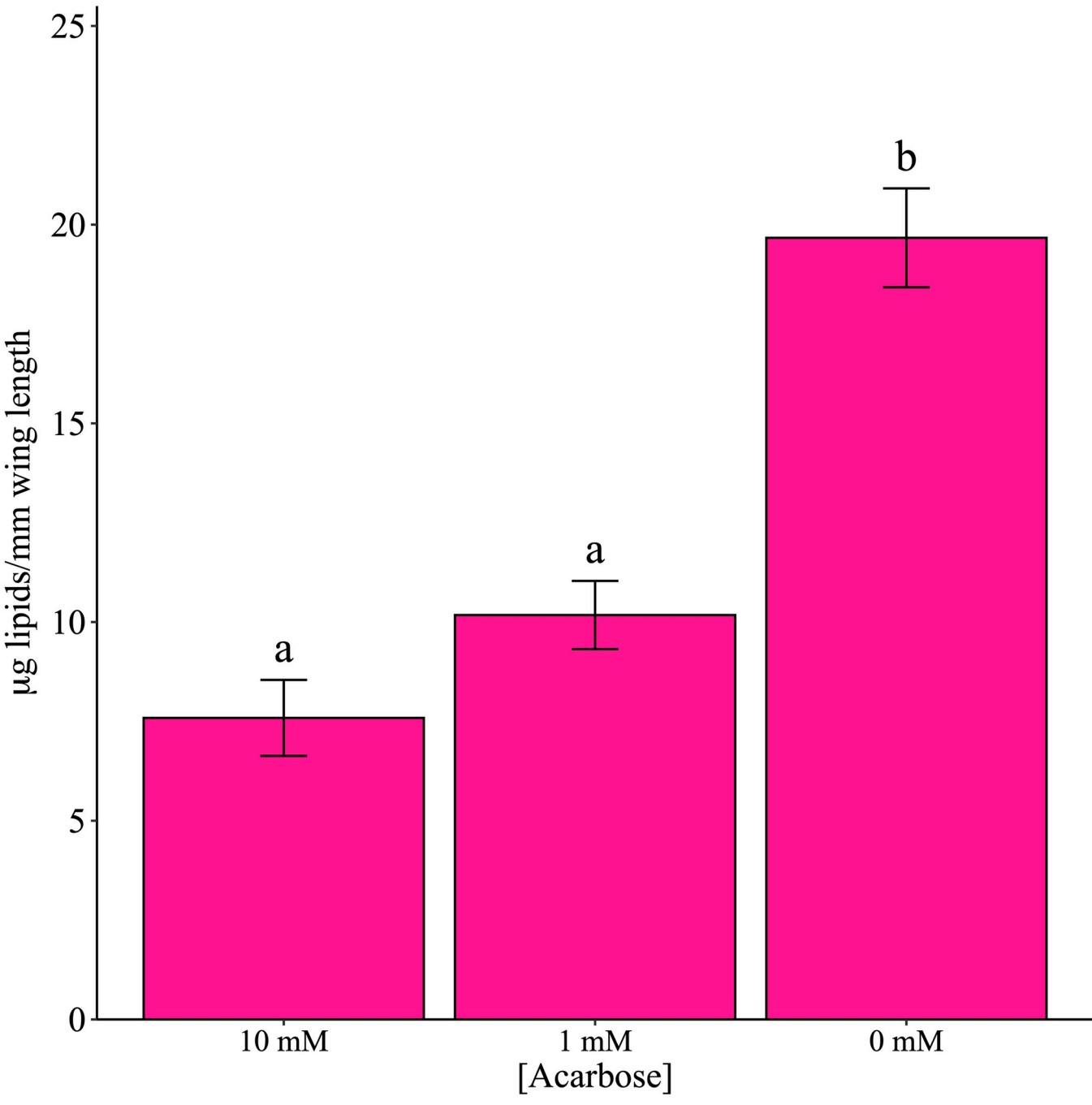

**Fig 5. Lipid measurements in adult *Culex quinquefasciatus* after no-choice exposure to acarbose using the vanillin method.** Lipid concentration was determined by pink color development. Different lowercase letters signify statistical significance of pairwise comparisons using Tukey's method at α = 0.05.

Trends in the expression of the putative glucosidase genes were not consistent with the observed enzyme activities. There are several reasons for lack of concordance between activity and gene expression. Glucosidase protein expression levels may not correlate well with activity due to differential degradation of the functional protein, i.e., that the steady state of these proteins did not reveal any trends in gene expression that could be generalized across life stages [45]. Second, it is also possible that our homology search to identify putative glucosidases did not identify the specific

transcripts responsible for the functional activity assessed in the biochemical assays. Glucosidases can sometimes have very specific substrate activity patterns and as we did not express these putative proteins, it is difficult to be sure that we have identified the correct transcripts. Instead, we feel that the utility of our expression data is to simply show that expression across life stages is dynamic like glucosidase activity levels. A follow-up study utilizing a multiomics approach would be useful in elucidating the steady state of these proteins and their associated transcripts.

Inhibition of α-glucosidases by acarbose in adult *Cx. quinquefasciatus* produced marked mortality both in females but especially in males, with clear reductions in *in vivo* activity, as well as in key energetic storage molecules. Specifically, the reduction in energetic storage molecules supports the likeliest explanation of mortality being starvation. It appears that there is a narrow threshold at which inhibition of α-glucosidases becomes lethal versus not. At 10 mM, acarbose produced around 60% mortality in females at 4 d, while mortality of males exceeded 90% at the same timepoint. It is interesting to see how sensitive males were in comparison to females. One explanation for this relatively large difference in response could have to do with typical adult diets. While females feed on both plant sugars and blood, males solely rely on plant sugars [32] and do not take blood meals. Because we assayed only females at the conclusion of the inhibition studies, it is unclear what impacts α-glucosidase inhibition had on *in vivo* activity and depletion of energetic storage molecules in males. This would be a worthwhile subject for a follow up study. Both glycogen and lipids are replenished from frequent sugar feeding by adult mosquitoes (reviewed in [3]). Upon emergence, adult mosquitoes quickly try to acquire a sugar meal to prolong flight. While we did not quantify flight activity during the inhibition experiments, mosquitoes in treatment arenas were noticeably less active than non-inhibited controls.

Predictably, inhibition of β-glucosidases did not yield mortality in larvae at 24 h. The larval diet consisted of crude fats, proteins, and carbohydrates that would require an array of enzyme classes to extract nutritional value. Interestingly, the known β-glucosidase inhibitor conduritol β-epoxide was a slightly better α-glucosidase inhibitor in *Cx. quinquefasciatus* larvae than it was a β-glucosidase inhibitor. Even so, the highest treatment concentration of 1 mM left 52.0% α-glucosidase activity and 72.1% β-glucosidase activity remaining after inhibition. Marked mortality was observed in the adult α-glucosidase inhibition assay when only around 20% activity remained. This suggests that the glycoside hydrolases in *Cx. quinquefasciatus* may have high rates of substrate turnover and thus can withstand a relatively high degree of impairment with few consequences, at least within 24 h.

Further study of glycoside hydrolases is of value in *Cx. quinquefasciatus* and other mosquitoes generally, because of their obligate sugar feeding behaviors (reviewed in [3]) and how this forces them to interact with their environment. Glycoside hydrolases, including α- and β-glucosidases, have been studied in numerous insects before with a similar theme [46–49]. Key processes like sugar digestion offer new insights into potential physiological weaknesses that may be exploitable for control or surveillance purposes in *Cx. quinquefasciatus*.

## Supporting information

**S1 File. Raw data for all experiments.**
(XLSX)

## Acknowledgments

We thank ME Scharf for conversations about glycoside hydrolases and the biochemical assays used here. We also thank M Mosore, OS Baker, and NN Barguez-Arias for assistance with the larval inhibition assays.

NDS and ASE are employees of the US Government, and this work was conducted as part of their official duties. Mention of trade names or commercial products in this publication is solely for the purpose of providing specific information and does not imply recommendation or endorsement by the U.S. Department of Agriculture. USDA is an equal opportunity provider and employer.

## Author Contributions

**Conceptualization:** Edwin R. Burgess, IV, Caitlin E. Taylor, Lyle J. Buss, Alden S. Estep.

**Data curation:** Edwin R. Burgess, IV, Caitlin E. Taylor, Lyle J. Buss, Alden S. Estep.

**Formal analysis:** Edwin R. Burgess, IV, Neil D. Sanscrainte, Alden S. Estep.

**Investigation:** Edwin R. Burgess, IV, Neil D. Sanscrainte, Caitlin E. Taylor, Lyle J. Buss, Alden S. Estep.

**Methodology:** Edwin R. Burgess, IV, Neil D. Sanscrainte, Caitlin E. Taylor, Lyle J. Buss, Alden S. Estep.

**Project administration:** Edwin R. Burgess, IV.

**Resources:** Edwin R. Burgess, IV, Neil D. Sanscrainte, Alden S. Estep.

**Visualization:** Edwin R. Burgess, IV.

**Writing – original draft:** Edwin R. Burgess, IV.

**Writing – review & editing:** Edwin R. Burgess, IV, Neil D. Sanscrainte, Caitlin E. Taylor, Lyle J. Buss, Alden S. Estep.

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
