## [Decision Letter · Decision Letter 0]

24 Jul 2023

PONE-D-23-15295Expression, activity, and consequences of biochemical inhibition of α- and β-glucosidases in different life stages of Culex quinquefasciatusPLOS ONE

Dear Dr. Burgess,

Thank you for submitting your manuscript to PLOS ONE. After careful consideration, we feel that it has merit but does not fully meet PLOS ONE’s publication criteria as it currently stands. Therefore, we invite you to submit a revised version of the manuscript that addresses the points raised during the review process.

We look forward to receiving your revised manuscript.

Kind regards,

Rachid Bouharroud

Academic Editor

PLOS ONE

Additional Editor Comments:

Dear Authors

The work deserve to be published but some minor comments should be addressed in order to achieve best results.

Regards

Reviewers' comments:

Reviewer's Responses to Questions

**Comments to the Author**

1. Is the manuscript technically sound, and do the data support the conclusions?

Reviewer #1: Yes

Reviewer #2: Yes

2. Has the statistical analysis been performed appropriately and rigorously? 

Reviewer #1: Yes

Reviewer #2: Yes

3. Have the authors made all data underlying the findings in their manuscript fully available?

Reviewer #1: Yes

Reviewer #2: Yes

4. Is the manuscript presented in an intelligible fashion and written in standard English?

Reviewer #1: Yes

Reviewer #2: No

5. Review Comments to the Author

Reviewer #1: The results reported by this study are original and have not been published elsewhere and suggest the importance of α- and β-glucosidases on the survival in the 3 life stages of Cx. Quinquefasciatus studied. The experiments were described with sufficient details and the statistical analysis were well performed. The conclusions were supported accordingly to the data; however, the discussion may need a fresh literature for more clarity about the research in parallel that were done during the last few years. Therefore, I recommend a minor correction.

Reviewer #2: The manuscript is interesting and provides valuable insights into the role of α-glucosidases and β-glucosidases in the survival of Cx. quinquefasciatus mosquitoes in different life stages. The findings contribute to our understanding of mosquito digestion, species-specific knowledge, biocontrol agents, and potential applications for vector control strategies.

However, I suggested an English review, I added some comments in the pdf. file.

6. PLOS authors have the option to publish the peer review history of their article (what does this mean?). If published, this will include your full peer review and any attached files.

Reviewer #1: No

Reviewer #2: **Yes: **Maísa da Silva Araújo

---

## [Author Response · Author response to Decision Letter 0]

31 Jul 2023

Response to reviewers:

We thank the reviewers for their helpful comments that have improved our manuscript. We have made comments below addressing the general review statement. Additional, Reviewer #2 sent along a document with grammatical comments. We have generated a Word document with Track Changes added to it documenting which changes we made and which we didn’t, along with an explanation for why we chose not to make the change. Comments below are preceded by ***.

Reviewer #1: The results reported by this study are original and have not been published elsewhere and suggest the importance of α- and β-glucosidases on the survival in the 3 life stages of Cx. Quinquefasciatus studied. The experiments were described with sufficient details and the statistical analysis were well performed. The conclusions were supported accordingly to the data; however, the discussion may need a fresh literature for more clarity about the research in parallel that were done during the last few years. Therefore, I recommend a minor correction.

***We thank this reviewer for the flattering words on our work. We were very unsure of what this reviewer means by “fresh literature” about the “research in parallel.” To our knowledge, and with extensive literature searches preceding the submission of this manuscript, we found very little recent work done on the topic of glucosidases in mosquitoes. Of the ones we found, we addressed them in the introduction. As it stands, we feel that the discussion addresses our discoveries based on what is known and accepted in the entomological literature on this topic. We will default to what the editor would like us to do, if anything, to address this comment.

Reviewer #2: The manuscript is interesting and provides valuable insights into the role of α-glucosidases and β-glucosidases in the survival of Cx. quinquefasciatus mosquitoes in different life stages. The findings contribute to our understanding of mosquito digestion, species-specific knowledge, biocontrol agents, and potential applications for vector control strategies.

However, I suggested an English review, I added some comments in the pdf. file.

***This reviewer provided a comprehensive review of our document’s grammar and syntax. We thank them for catching some errors and in general striving to improve the readability and flow of our manuscript. We have made a good number of the requested changes, while leaving others unchanged. We have added our changes and response to the comments in a Track Changes version of our manuscript.

---

## [Editor Report · Decision Letter 1]

9 Aug 2023

Expression, activity, and consequences of biochemical inhibition of α- and β-glucosidases in different life stages of Culex quinquefasciatus

PONE-D-23-15295R1

Dear Dr. Burgess,

We’re pleased to inform you that your manuscript has been judged scientifically suitable for publication and will be formally accepted for publication once it meets all outstanding technical requirements.

Kind regards,

Rachid Bouharroud

Academic Editor

PLOS ONE

Additional Editor Comments (optional):

Dear

Happy for the acceptance of this valuable work related to public health. Your manuscript looks currently good to be published by PlosOne.

Regards
---

## [Editor Report · Acceptance letter]

21 Aug 2023

PONE-D-23-15295R1 

Expression, activity, and consequences of biochemical inhibition of a- and b-glucosidases in different life stages of *Culex quinquefasciatus*

Dear Dr. Burgess:

I'm pleased to inform you that your manuscript has been deemed suitable for publication in PLOS ONE. Congratulations! Your manuscript is now with our production department. 

Kind regards, 

on behalf of

Dr. Rachid Bouharroud 

Academic Editor

PLOS ONE